# Inhibitory Effect of Catechin-Rich Açaí Seed Extract on LPS-Stimulated RAW 264.7 Cells and Carrageenan-Induced Paw Edema

**DOI:** 10.3390/foods10051014

**Published:** 2021-05-06

**Authors:** Gabriel Silva Xavier, Amanda Mara Teles, Carla Junqueira Moragas-Tellis, Maria do Socorro dos Santos Chagas, Maria Dutra Behrens, Wendel Fragoso de Freitas Moreira, Ana Lucia Abreu-Silva, Kátia da Silva Calabrese, Maria do Desterro Soares Brandão Nascimento, Fernando Almeida-Souza

**Affiliations:** 1Rede Nordeste de Biotecnologia, Universidade Federal do Maranhão, São Luís 65080-805, Maranhão, Brazil; xaviersilva.g@hotmail.com; 2Pós-Graduação em Saúde do Adulto, Universidade Federal do Maranhão, São Luís 65080-805, Brazil; damarateles@hotmail.com; 3Laboratório de Produtos Naturais para Saúde Pública (LPNSP), Instituto de Tecnologia em Fármacos—Farmanguinhos, Fiocruz, Rio de Janeiro 21041-000, Brazil; carla.tellis@far.fiocruz.br (C.J.M.-T.); msocchagas@gmail.com (M.d.S.d.S.C.); mariabehrens@hotmail.com (M.D.B.); 4Pós-graduação em Ciência Animal, Universidade Estadual do Maranhão, São Luís 65055-310, Maranhão, Brazil; wendelmoreira1997@hotmail.com (W.F.d.F.M.); abreusilva.ana@gmail.com (A.L.A.-S.); fernandoalsouza@gmail.com (F.A.-S.); 5Laboratório de Imunomodulação e Protozoologia, Instituto Oswaldo Cruz, Fiocruz, Rio de Janeiro 21040-900, Brazil

**Keywords:** inflammation, *Euterpe oleracea*, fruit, nitric oxide, cytokines, histology, mast cells

## Abstract

Açaí berry is a fruit from the tree commonly known as açaízeiro (*Euterpe oleracea* Mart.) originated from the Amazonian region and widely consumed in Brazil. There are several reports of the anti-inflammatory activity of its pulp and few data about the seed’s potential in inflammation control. This work aimed to evaluate the effect of catechin-rich açaí extract on lipopolysaccharide (LPS)-stimulated RAW 264.7 cells and carrageenan-induced paw edema. The treatment with *E. oleracea* ethyl acetate extract (EO-ACET) was used in an in vitro model performed with macrophages stimulated by LPS, in which pro-inflammatory markers were evaluated, and in an in vivo model of acute inflammation, in which edema inhibition was evaluated. EO-ACET showed an absence of endotoxins, and did not display cytotoxic effects in RAW 264.7 cells. LPS-stimulated cells treated with EO-ACET displayed low levels of nitrite and interleukins (IL’s), IL-1β, IL-6 and IL-12, when compared to untreated cells. EO-ACET treatment was able to inhibit carrageenan-induced paw edema at 500 and 1000 mg/kg, in which no acute inflammatory reaction or low mast cell counts were observed by histology at the site of inoculation of λ-carrageenan. These findings provide more evidence to support further studies with *E. oleracea* seeds for the treatment of inflammation.

## 1. Introduction

The use of dietary supplements of botanical origin has been widespread worldwide and the advancement of studies in this area has been possible due to the involvement of the National Center for Complementary and Alternative Medicine, the Office of Dietary Supplements, as well as other components of the National Institute of Health. The use of these supplements has health, as well as economic, implications [1,2]. However, it is very important to draw attention to the veracity of information contained on product labels, which may be configured if it does not comply with the uniformity test of dosage forms. Therefore, many studies described in the literature show the need for the better quality control of these products and the existence of stricter rules [3].

Compounds from plant foods are cheap sources of products with anti-inflammatory and antioxidant properties. The increasing interest of consumers in foods with high health potential made the pomace from berries a good source of bioactive compounds [4]. These studies seem to look for compounds that can simultaneously inhibit cyclooxygenase-2 (COX-2) and 5-lipoxygenase (5-LOX), which are called double COX/5-LOX inhibitors. In this field, anthocyanins are identified as compounds that have anti-inflammatory activity. Various processed fruit products have traditionally been used to treat colds and flu [5].

*Euterpe oleracea* Mart. belongs to the Arecaceae family and is commonly known as “açaizeiro”. It is the most popular large palm tree (Figure 1A) from the floodplain Amazon region, measuring from 6 to 14 m in height. The fruit is a small berry, with a size that varies between 1 to 2 cm in diameter and weighs an average of 1.2 g (Figure 1B). The epicarp (bark) and the mesocarp (pulp) is dark purple when ripe. Mesocarp measures approximately 1–2 mm thick and surrounds the endocarp, which is a bulky and hard structure, whose shape follows that of the fruit, which contains a seed. The açaí seed fills most of the fruit, representing 73% of its mass, presenting a globose shape, dark brown color and an average diameter of 11.5 mm (Figure 1C) [6,7,8,9].

Different parts of *E. oleracea* have been studied for chemical composition and pharmacological properties [10]. The oil extracted from the pulp or seed of the fruit presents mainly fatty acids, whose oleic acid (47.58%), palmitic (24.06%) and linoleic (13.58%) acids are the major compounds, while palmitoleic, vaccenic, lauric and stearic acids were minor constituents [11]. High concentrations of (+)-catechin and procyanidin oligomers have also been found in *E. oleracea* pulp and oil extracts [12].

The fruit açaí, is widely consumed by the population in Brazil [13]. Besides the nutritional value, açai berry presents antioxidant properties due to its high level of phenolic compounds, such as anthocyanins [14], which also exerts anti-inflammatory, anti-proliferative and cardioprotective activities [15]. A variety of phenolic compounds identified in *E. oleracea*, such as the flavonoids quercetin, vitexin, luteolin, chrysoeriol and dihydrocaempferol, and the anthocyanins cyanidin-3-rutinoside, cyanidin-3-glucoside, cyanidin-3-sambioside and peonidine-3-rutoside, showed antioxidant effect [16].

The literature shows that both the pulp and the ethanolic extract of the fruit have anti-inflammatory action. It was demonstrated that C57BL/6 mice exposed to cigarette smoke and treated with hydroalcoholic fruit extract showed a lower number of neutrophils and macrophages in the lung when compared to the control group, which demonstrates the anti-inflammatory activity of the açaí fruit extract [17]. Other experiments showed that both dried and frozen hydroalcoholic extract of the fruit pulp interfered with the activation and proliferation of macrophages, promoting the arrest of the cell cycle due to the reduction of NOD-like receptor family pyrin domain containing protein 3 (NLRP3) activation [18].

The anti-inflammatory activity of açaí is often associated with the presence of flavonoids, such as catechin and epicatechin, both in the pulp fruit [18] and in the seeds [19]. Catechin’s anti-inflammatory effect is well described alone, acting via the regulation of Toll-like receptor (TLR) 2/4 and inflammasome signaling [20], or in combination with other compounds, suppressing the production of inflammatory cytokines in mouse macrophages in vitro in association with baicalin and β-caryophyllene [21], and inhibiting the activation of TLR4-MyD88-mediated nuclear factor kappa-light-chain-enhancer of activated B cells (NF-κB) and mitogen activated protein kinase (MAPK) signaling pathways in RAW 264.7 macrophages in association with quercetin [22].

The use of digested or undigested açaí seed extract, rich in catechin and epicatechin, shows that there was a reduction in the activation of NF-κB and levels of tumor necrosis factor alpha (TNF-α), confirming that the anti-inflammatory activity of the extract is not altered after ingestion, which suggests the açaí seeds may be used to obtain and bioactive compounds for various purposes [19]. However, besides the use of the in vitro digested açaí seed extract, the study only performed in vitro experiments, which limits the evidence of the possible anti-inflammatory potential of the açaí seed. Thus, this work aimed to evaluate the effect of catechin-rich açaí extract on lipopolysaccharide (LPS)-stimulated RAW 264.7 cells and the in vivo model of acute inflammation carrageenan-induced paw edema.

## 2. Materials and Methods

### 2.1. Plant Material

The fruits of *E. oleracea* used in this study came from Parque da Juçara, São Luís, Maranhão, Brazil (latitude −2.6274201, longitude −44.2922708). The collection was carried out in the dry period in September 2017. A sample of the specimen was stored under exsiccate number 30, issued by Herbário Rosa Mochel of the Nucleus of Biological Studies of the State University of Maranhão (UEMA). To obtain the seeds, the fruits were washed in running water and in distilled water at 60 °C for 5 min, and the pulp was removed with the aid of a pulper. The seeds were placed to dry in an oven at 47 °C. After drying, the fibers covering the seed were removed manually.

### 2.2. Obtaining of Ethyl Acetate Extract from E. oleracea Seed

The dried seeds were ground in a manual mill to facilitate the extraction process. Then, 500 g of the crushed açaí seeds were weighed and added to 400 mL of 70% hydroethanolic solution. The mixture remained under constant agitation for 2 h and was then left to rest for extraction by maceration, with four successive changes of solvent, for four days. After the extraction process, the resulting extract was filtered on Whatman filter paper #1, and the solvent concentrated on a low-pressure rotary evaporator (Fisatom Equipamentos Científicos Ltd.a., São Paulo, Brazil) at 40 °C. The hydroethanolic extract was then lyophilized (LIOTOP model 202, Fisatom Equipamentos Científicos Ltd.a., São Paulo, Brazil) at a temperature of −30 to −40 °C and a vacuum of 200 mm Hg. *E. oleracea* hydroethanolic extract (EO-HE), 15 g, was diluted in 200 mL of methanol:water (8:2; *v*/*v*) solution and partitioned in a liquid separation funnel, using solvents of increasing polarity (chloroform, ethyl acetate and water). The partition resulted in fractions named EO-CLO, EO-ACET, and EO-AQ, respectively. All the fractions were subjected to evaporation under reduced pressure to obtain dry matter and were then identified and kept at −20 °C. Before the biological assays, the fractions were solubilized in dimethyl sulfoxide (DMSO) (Sigma-Aldrich, St Louis, MO, USA) at 100× the final concentration for in vitro assays, and the final test concentrations in Dulbecco’s Modified Eagle Medium (DMEM) culture medium (Sigma-Aldrich, St Louis, MO, USA) presented less than 0.5% DMSO. For in vivo assays, PBS was used to solubilize the EO-ACET to the final doses. Dilution extracts were prepared immediately before use.

### 2.3. Thin Layer Chromatography (TLC) Analysis

The hydroalcoholic extract and fractions were preliminarily analyzed by Thin Layer Chromatography (TLC) using silica gel 60 F254 in an aluminum chromatography sheet (20 cm × 20 cm × 0.15 mm; Merck, Darmstadt, Germany) previously activated in an oven at 105 °C for 2 h. As a mobile phase, the mixture of solvents methanol: chloroform: formic acid (8.5:1.5:0.5 *v*/*v*/*v*) was used. After elution, chromatograms were evaluated under visible and ultraviolet light at wavelengths of 254 and 365 nm, followed by spraying with sulfuric vanillin reagent (1% in methanol) and subsequent heating at 110 °C for 2 min.

### 2.4. Analysis by High-Performance Liquid Chromatography Coupled to Diode-Array Detection and Mass Spectrometry (HPLC-DAD-MS)

The HPLC analysis was carried out with a modified C18 column 250 mm × 4.6 mm × 5 µm (Shim-pack CLCODS, Shimadzu, Canby, Oregon). The solvents used were (A) water acidified with 5% formic acid and (B) methanol HPLC grade. The elution gradient established was 15% B for 5 min, 15 to 80% B in 25 min, and maintaining 80% B isocratic for 15 min to rebalance the column, using a flow rate of 1.0 mL/min. The mass detection was performed in a positive mode with a capillary voltage of 2500 V; end plate offset: 2000 V; capillary output 110 V, skimmer 1 20 V, skimmer 2 10 V, dry gas (N_2_) temperature 325 °C and flow 11 L/min, nebulizer 60 psi, sweep range from 200 to 800 *m*/*z* temperature set at 25 °C. Dual online DAD detection was performed using 280 and 520 nm as the wavelengths of choice.

### 2.5. Quantification of Endotoxins

The quantification of endotoxin in the EO-ACET dilutions (125, 250 and 500 µg/mL) was carried out following the recommendations of Pierce™ LAL Chromogenic Endotoxin Quantitation Kit (Thermo Scientific, Carlsbad, CA, USA) [23].

### 2.6. Cell Culture

RAW 264.7 cell line murine macrophages (ATCC^®^ TIB-71™) were maintained in DMEM supplemented with 10% fetal bovine serum (FBS) (Gibco, Gaithersburg, MD, USA), penicillin (100 U/mL) and streptomycin (100 μg/mL) (Sigma-Aldrich, St Louis, MO, USA) at 37 °C and 5% CO_2_ in culture flasks.

### 2.7. Cytotoxicity Assay

RAW 264.7 cells (2 × 10^6^ cells/mL, 100 µL per well) were incubated overnight in 96 well plates for adhesion. The medium and non-adherent cells were removed, and the adherent cells were treated with 100 µL of different concentrations of EO-ACET (125, 250 and 500 µg/mL), diluted in DMEM. Wells without cells only with the medium were used as blanks, and wells with cells and DMSO 1% were used as a control. Wells with cells treated with LPS (Sigma-Aldrich, St Louis, MO, USA) at 10 µg/mL and with dexamethasone (Sigma-Aldrich, St Louis, MO, USA) at 100 µM. After 48 h of treatment, cell viability assay was performed using the colorimetric MTT [3-(4,5-dimethylthiazol-2-yl)-2,5-diphenyl tetrazolium bromide] (Sigma-Aldrich, St Louis, MO, USA) method [24], with modifications [25]. Briefly, 10 µL of MTT at 5 mg/mL was added and incubated for two hours at 37 °C and 5% CO_2_. The supernatants were removed, and the formazan crystals were solubilized with 100 µL of DMSO. The absorbance was obtained with a spectrophotometer at 540 nm wavelength. Cytotoxicity was expressed as a percentage, as described elsewhere [26].

### 2.8. EO-ACET Treatment in RAW 264.7 Macrophages Stimulated with LPS

RAW 264.7 macrophages (2 × 10^6^ cells/mL) were incubated in 24-well plates overnight. After the removal of non-adherent cells, the adherent cells were treated with EO-ACET at 125, 250 and 500 µg/mL concentrations, or with dexamethasone (100 µM) for one hour, and then stimulated, or not, with LPS (10 μg/mL). Cells non-treated and non-stimulated, and cells non-treated and stimulated were used as controls. All treatment dilutions were carried out in DMEM medium. After 48 h, the supernatant was collected for the quantification of nitrite and cytokines interleukin (IL) 1 beta (IL-1β), IL-6 and IL-12.

### 2.9. Nitrite and Cytokines Quantification

Nitrite levels in cell culture supernatant were determined by the Griess method [27]. Then, 50 µL of supernatant was added to 50 µL of Griess reagent (25 µL of sulfanilamide 1% in 2.5% H_3_PO_4_ solution and 25 µL of N-(1-naphthyl) ethylenediamine 0.1% solution) in 96-well plates. After 10 min protected from light, the absorbance of the plate was measured in a spectrophotometer at 540 nm. The results were expressed in NaNO_2_ (µM), based on a standard curve with known concentrations of sodium nitrite (Sigma-Aldrich, St Louis, MO, USA), at 100 to 3.1 µM NaNO_2_, obtained for serial dilution 1:2 [28]. The cytokine quantification of IL-1β, IL-12 and IL-6 (BD OptEIA™) was performed following the manufacturer’s specifications.

### 2.10. Animals and Ethical Statement

Female BALB/c mice from six to eight weeks of age were obtained from the Institute of Science and Technology in Biomodels (ICTB/FIOCRUZ) and maintained under pathogen-free conditions at a controlled temperature, with food and water ad libitum. The experiments were conducted in accordance with the National Council for the Control of Animal Experimentation (CONCEA) and approved by the Ethics Commission for the Use of Animals of the Oswaldo Cruz Institute, license number CEUA/IOC—L053/2016, 28 December 2016.

### 2.11. Paw Edema Induced by λ-Carrageenan

The paw edema was carried out as described by Oliveira et al. (2019) [29]. Mice were separated into six groups of five animals. Four groups were pre-treated with 250, 500 or 1000 mg/kg of EO-ACET by gavage, or dexamethasone (5 mg/kg solubilized in PBS, intramuscular route), and two groups were pre-treated with PBS by gavage. After one hour, 25 µL of λ-carrageenan 1% was injected into the left hind footpad. The control group was inoculated with PBS and pre-treated with PBS. After 1, 2, 3 and 4 h of λ-carrageenan inoculation, footpad swelling was measured using a Schnelltaster dial gauge caliper (Kröplin GmbH, Hessen, Germany). The edema thickness was expressed in millimeters and obtained from the difference between the inoculated footpad at the evaluation time and its own basal level at time 0 before inoculation. The animals were euthanized, four hours after λ-carrageenan inoculation, with 250 μL intraperitoneal injection of a 1:1 mixture of ketamine (100 mg/mL; Syntec, BRA) and xylazine (20 mg/mL; Syntec, BRA). Fragments of footpad edema were collected for histological analysis.

### 2.12. Histology Analysis

Skin fragments from footpad were fixed in 10% buffered formalin and routinely processed for paraffin embedding. Tissue sections at 5 μm thick were stained with Hematoxylin-Eosin (HE) and Giemsa using the modified Wolbach method for the quantification of mast cells. Tissues were observed, analyzed by a researcher with expertise, blinded to the experimental groups. In HE staining, the inflammatory parameters analyzed were edema, congestion and inflammatory infiltrate. Mast cells counting were performed under a light microscope and representative areas in five fields were selected [29].

### 2.13. Statistical Analysis

The results are represented as mean ± standard deviation and were analyzed by Kruskal–Wallis followed by Dunn’s multiple comparisons tests (*p* < 0.05) or by analysis of variance (two-way ANOVA) followed by Bonferroni’s multiple comparisons test (*p* < 0.05). Statistical analyses were carried out with GraphPad Prism 7.00 software package (GraphPad Software, San Diego, CA, USA).

## 3. Results

### 3.1. Chemical Characterization of E. oleracea Seed Extracts

Thin-layer chromatography with catechin standard showed the presence of catechin in *E. oleraceae* hydroalcoholic extract (EO-HE) and ethyl acetate fraction (EO-ACET), slight evidence in chloroform fraction (EO-CLO) and absence in aqueous fraction (EO-AQ) (Figure 2A). The HPLC-MS analysis confirmed this result, as observed in the four chromatograms shown in Figure 2B. The signs found in each chromatogram, with their respective relative percentage areas and their tentative identification, are described in Table 1.

### 3.2. EO-ACET Is Endotoxin-Free and Not Displayed Cytotoxicity

The endotoxin quantification shows endotoxin absence in all EO-ACET dilutions used in macrophage treatment. Cytotoxicity assay performed with the same treatment conditions of posterior experiments did not show significant alteration in the percentage of viable cells (Figure 3).

### 3.3. EO-ACET Reduced Levels of Pro-Inflammatory Markers in RAW 264.7 Cells Stimulated with LPS

The treatment of RAW 264.7 macrophages without LPS stimulation with EO-ACET at the highest concentration analyzed, 500 μg/mL, did not alter the levels of nitrite or cytokines in the cell culture supernatant. However, LPS-stimulated cells treated with EO-ACET showed low levels of all four pro-inflammatory markers analyzed when compared to untreated and LPS-stimulated cells. Treatment with EO-ACET at 125 and 500 μg/mL displayed low levels of nitrite (*p* = 0.0237 and *p* = 0.0028, respectively; Figure 4A), while only at 500 μg/mL did they present low levels of IL-1β (*p* = 0.0058; Figure 4B). The inhibitory effect of EO-ACET was more evident in IL-6 (Figure 4C) and IL-12 quantification (Figure 4D). All the three concentrations analyzed (125, 250 and 500 μg/mL) showed low levels of IL-6 (*p* < 0.0001, *p* = 0.0002 and *p* = 0.0166, respectively) and IL-12 (*p* = 0.0194, *p* = 0.0059 and *p* = 0.0257, respectively). The reference drug dexamethasone treatment presented low levels of nitrite (*p* = 0.0001) and cytokines IL-1β (*p* = 0.0093), IL-6 (*p* < 0.0001) and IL-12 (*p* = 0.0002), as expected (Figure 4).

### 3.4. EO-ACET Inhibited Paw Edema Induced by λ-Carrageenan

The treatment with EO-ACET demonstrated lower edema thickness than the PBS group four hours after λ-carrageenan inoculation (Figure 5A and Table 2). After four hours, EO-ACET treatment inhibited edema in a dose-dependent manner at 500 and 1000 mg/kg (Figure 5B). The histological analysis showed that the animals treated with 1000 mg/kg presented the same results as the animal treated with dexamethasone; in other words, no acute inflammatory reaction was observed at the site of inoculation of carrageenan, as evidenced by histology of mast cells (Figure 6). Regarding the participation of the mast cells in the process, the groups treated with EO-ACET at 500 and 1000 mg/kg presented lower mean mast cell numbers in comparison with the untreated and stimulated group (Figure 7).

## 4. Discussion

The study demonstrated the anti-inflammatory properties of the ethyl acetate fraction from *E. oleracea* used in Brazil. First, it was performed a chemical characterization of the extract and their fractions obtained with chloroform, ethyl acetate, and water to define the catechin-rich one to be used in the biological experiments. The phytochemical analysis showed that in the ethyl acetate fraction, mainly catechins and pelargonidin were identified, which differ from the previous results of our laboratory, in which a predominance of proanthocyanidin A2, trimeric and tetrameric procyanidins was observed [30]. Other studies also describe the occurrence of catechins and epicatechin in the açaí seed [31,32].

Although the seeds of the present study were collected in the same place and the same season as Freitas et al. [30], several factors may have interfered, such as the edaphoclimatic conditions, rainfall index and, mainly, the storage time that in the present work was not longer. Among the factors that cause the instability of anthocyanins are pH, copigmentation, light, temperature, metals and oxygen [33,34].

In an overview of food, therapeutic and industrial applications of Brazilian fruits of Arecaceae family, de Souza et al. considered açaí, from the physicochemical perspective, as a good supplement in the human diet due to its high content of compounds with known pharmacological properties and/or health benefits [35], such as oleic acid [36], anthocyanins [37], carotenoids [38] and phenolic compounds [39]. These compounds are related to the powerful antioxidant activity of berries [40] and other underused wild plants seeds [41] such as the acai seed. The “superfruit” nomination given to açaí fruit is due to the variety of bioactive compounds and their antioxidant ability.

Qualitative analysis by TLC revealed that EO-HE and EO-ACET displayed the presence of catechin, and the relative percentage area obtained by HPLC-DAD-MS confirmed this result, with EO-ACET exhibiting the higher amount of catechin and epicatechin between the analyzed fractions. Thus, EO-ACET was used in further experiments.

Before the in vitro treatment of RAW 264.7 cells, EO-ACET was subjected to an evaluation of the endotoxin quantity. Endotoxins induce pro-inflammatory effects [42] that would alter the in vitro and in vivo evaluation of the anti-inflammatory activity of *E. oleracea* seed extract. In addition, repeated exposure to LPS may induce a state of tolerance that reprograms the inflammatory response, resulting in reduced inflammatory cytokine production in vitro and in vivo [43] that would also modify EO-ACET anti-inflammatory response. The absence observed in this endotoxin analysis ensures that the fraction is free of contamination and that all the posterior experiments made with it had no interference of endotoxins.

The RAW 264.7 murine macrophages were used to assess anti-inflammatory activity, since they consist of a good in vitro model for the inhibition of the pathways that lead to the induction and production of pro-inflammatory enzymes and cytokines [29,44]. The lower levels of nitrite and cytokines observed in the supernatant of cells treated with EO-ACET indicates an inhibitory effect in the production of pro-inflammatory marker nitrite, IL-1β, IL-6 and IL-12 by macrophages.

Nitrite was used as an indirect way to quantify the amount of nitric oxide (NO) in culture supernatants. The NO originated from the conversion of L-arginine to L-citrulline by the NO synthase (NOS) enzyme that has three isoforms—euronal (nNOS); endothelial NOS (eNOS); inducible NOS (iNOS)—which is induced by pro-inflammatory factors such as cytokines or endotoxins and is widely expressed in macrophages [23,45]. The lower quantification observed in RAW 264.7 cells treated with EO-ACET revealed its inhibitory effect in NO production, as well as in the cytokines production.

The IL-1β is typically activated in macrophages after inflammasome sensing of infection or danger, leading to caspase-1 processing and driving inflammation after the release from macrophages [46]. The TLR4 receptor in monocytes and macrophages, when activated by LPS, induces IL-12 production and stimulates macrophages to produce other pro-inflammatory cytokines, driving inflammation [47]. On the other hand, IL-6 may have contextual protective or exacerbating roles during inflammation [48]. The inhibitory in vitro effect of EO-ACET treatment in IL-1β, IL-6 and IL-12 production by RAW 264.7 cells suggests a potential anti-inflammatory in vivo.

The anti-inflammatory activity of *E. oleracea* was also verified by a RAW 264.7 murine macrophage model using phytohemagglutinin. Açaí extract at 1 μg/mL modulated redox status by decreasing NLRP3 inflammasome levels and reducing pro-inflammatory cytokines IL-1 beta, IL-6, TNF-α, and IFN-ɣ, and the decreasing production of anti-inflammatory cytokine IL-10 [18]. In addition, the anti-inflammatory and antihypertensive effects of the açaí seed extract, rich in phenolic compounds, were attributed to the modulating of redox status by the positive modulation of the nuclear factor erythroid 2-related factor 2 (Nrf2) signaling pathway in human endothelial cells (HUVEC) [49].

In vitro experiments are important tools to evaluate a specific situation and control variables that cannot be controlled in vivo, and to provide insight into the mechanism of action of drugs candidate. However, in vitro studies also have their limitations in research involving inflammation. Inflammation is a complex process, involving several cells, biochemical mediators, and signaling molecules, that is not possible to mimic in vitro. In addition, is difficult to reproduce the same in vitro results in vivo due to pharmacokinetics and pharmacodynamics parameters [29,50]. Thereby, we carried out an in vivo acute inflammation model of carrageenan-induced paw edema in BALB/c mice to evaluate the anti-inflammatory activity of ethyl acetate fraction obtained from *E. oleracea* seed extract.

The animals treated with EO-ACET demonstrated edema inhibition, noticed by the lower increase in the thickness footpad when compared to untreated animals. The inhibitory activity was also higher in the highest doses, suggesting an ascending anti-inflammatory dose response. The edema event happens due to vascular alterations that initiate with the transitory constriction of the small vessels and posterior vasodilatation [51], generating deregulation of the osmotic balance. Furthermore, vasodilatation also allows leukocyte migration to the inflammatory site, producing additional inflammatory factors that sustain and potentiate the inflammation.

The in vivo anti-inflammatory activity of the *E. oleracea* seed extract was also observed in an experiment with C57BL/6 mice, in which açaí seed extract supplementation protected from obesity-associated hepatic steatosis and fibrosis, reducing oxidative stress, NF-κB expression and pro-inflammatory markers IL-6 and TNF-α [52].

Mast cells are one of the leucocytes that can migrate through vasodilatation, integrating a critical first line of defense against endogenous and environmental threats, and being intrinsically involved in the pathogenesis of skin inflammation. They are found mainly in vascularized tissues and have cytoplasmic granules that are rich in a wide variety of mediators as histamine, prostaglandins, heparin and serotonin [53,54]. We observed a decrease in the mast cell count in the footpads of animals treated with EO-ACET at the same treatment doses, which also inhibited edema increase, suggesting that mast cells may be involved in edema inhibition caused by EO-ACET treatment.

## 5. Conclusions

Overall, the results observed in vitro, with the inhibition of pro-inflammatory markers suggesting an anti-inflammatory activity of *E. oleracea* seed, was confirmed by the inhibition of carrageenan-induced paw edema induced by EO-ACET treatment. These findings add to the other descriptions of the inflammatory activity of açaí pulp fruit, and our results confirmed that the *E. oleracea* seed, which is rich in catechins, has anti-inflammatory potential. This study provides more evidence to support further studies with *E. oleracea* seeds for the treatment of inflammation.

## Figures and Tables

**Figure 1 foods-10-01014-f001:**
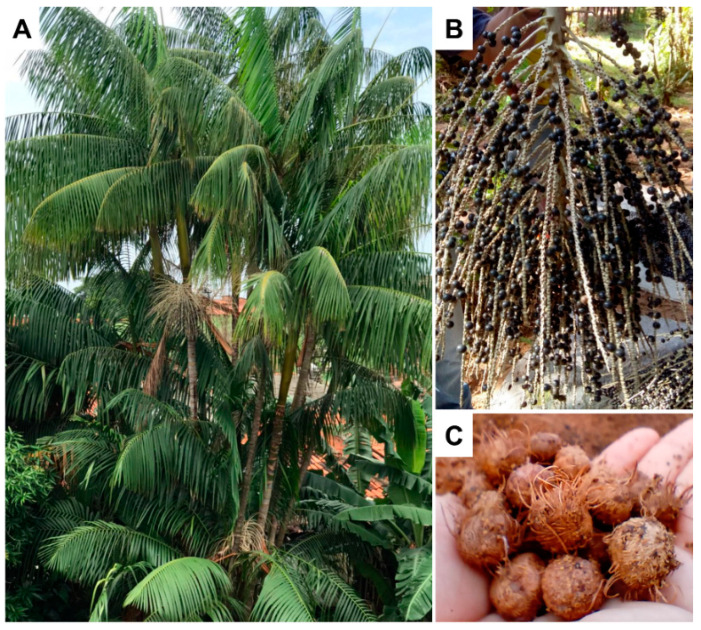
*Euterpe oleracea* tree (**A**), fruits (**B**) and seeds (**C**).

**Figure 2 foods-10-01014-f002:**
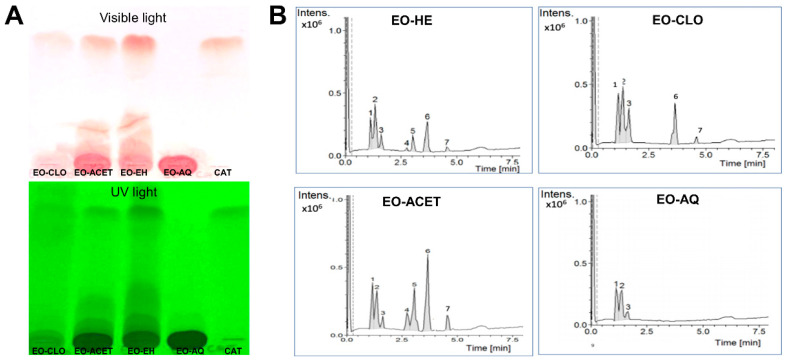
Chromatograms obtained by thin-layer chromatography (**A**) and high-performance liquid chromatography coupled to the mass spectrometer (**B**) of the hydroalcoholic extract of *Euterpe oleracea* seeds (EO-HE) and fractions obtained using chloroform (EO-CLO), ethyl acetate (EO-ACET) and water (EO-AQ), showing the presence of seven main signs.

**Figure 3 foods-10-01014-f003:**
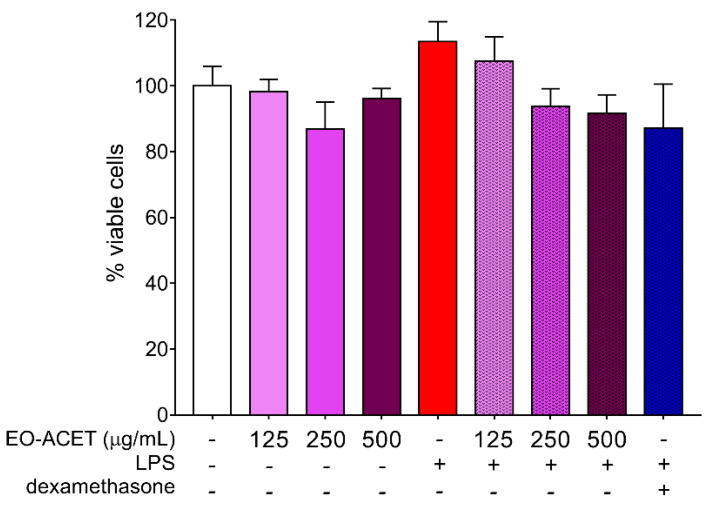
Cytotoxicity of *Euterpe oleracea* ethyl acetate fraction (EO-ACET) in RAW 264.7 cells after 48 h of treatment with EO-ACET (purple columns) or dexamethasone (blue column) at 100 μM and stimulated, or not, with LPS (red column) at 10 μg/mL. Data represent the mean ± standard deviation of the experiment performed in sextuplicate.

**Figure 4 foods-10-01014-f004:**
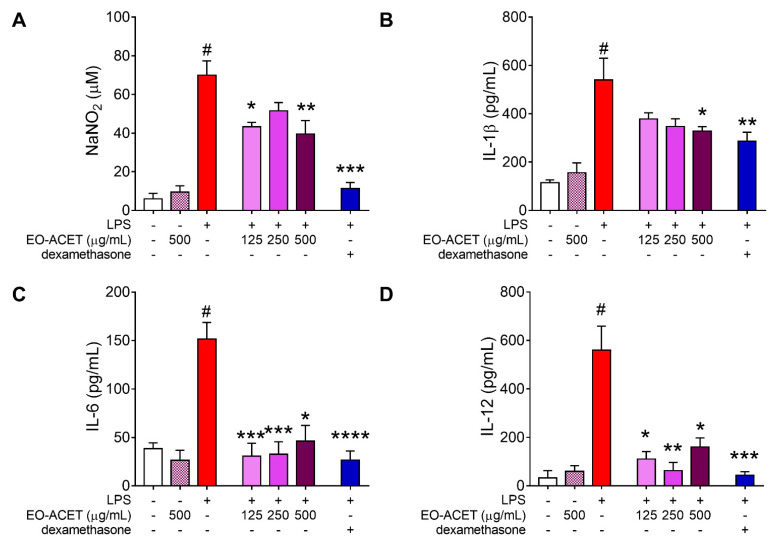
Pro-inflammatory factors in RAW 264.7 cells stimulated by LPS and treated with *Euterpe oleracea* ethyl acetate fraction (EO-ACET). Levels of nitrite (**A**) and cytokines (**B**–**D**) in culture supernatants of cells treated with EO-ACET (purple columns) or dexamethasone (blue column), 100 μM. Data represent the mean ± standard deviation of the experiment performed at least in triplicate. # *p* < 0.001 compared with the group without stimulation or treatment (white column); * *p* < 0.05, ** *p* < 0.01, *** *p* < 0.001, **** *p* < 0.0001 when compared with the stimulated and untreated group (red column) after Kruskal–Wallis analysis, followed by Dunn’s multiple comparisons tests.

**Figure 5 foods-10-01014-f005:**
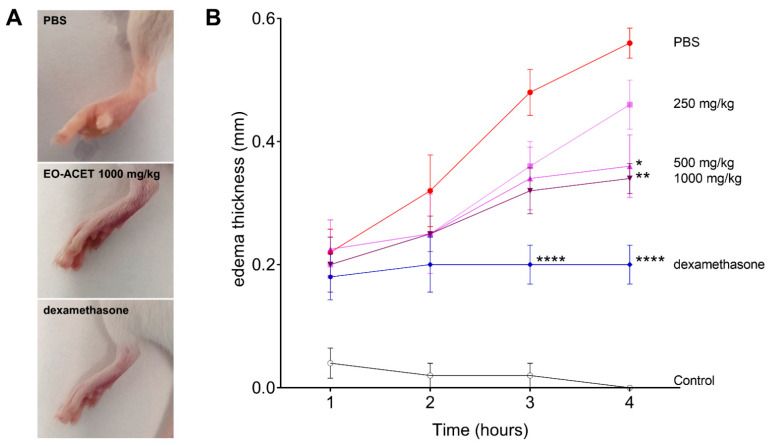
Paw edema of BALB/c mice inoculated with of λ-carrageenan and treated with *Euterpe oleraceae* ethyl acetate fraction (EO-ACET). Macroscopy (**A**) and kinetic of edema thickness (**B**) of animals inoculated with 25 μL of λ-carrageenan 1% and treated with 100 μL EO-ACET (purple lines) at 250, 500 or 1000 mg/kg by gavage or with dexamethasone (blue line) at 5 mg/kg via the intramuscular route. Data represent the experiment carried out in quintuplicate. * *p* < 0.05, ** *p* < 0.01, **** *p* < 0.0001, when compared with PBS group (red line), after analysis of variance (two-way ANOVA) followed by Bonferroni’s multiple comparisons tests.

**Figure 6 foods-10-01014-f006:**
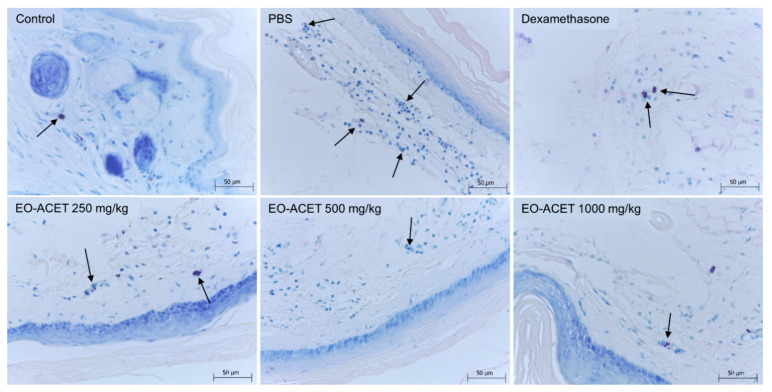
Mast cells (arrows) histology of BALB/c mice inoculated with of λ-carrageenan and treated with *Euterpe oleraceae* ethyl acetate fraction (EO-ACET). Animals inoculated with 25 μL of λ-carrageenan 1% and treated with 100 μL EO-ACET by gavage or with dexamethasone 5 mg/kg via the intramuscular route. Wolbach Giemsa Stain.

**Figure 7 foods-10-01014-f007:**
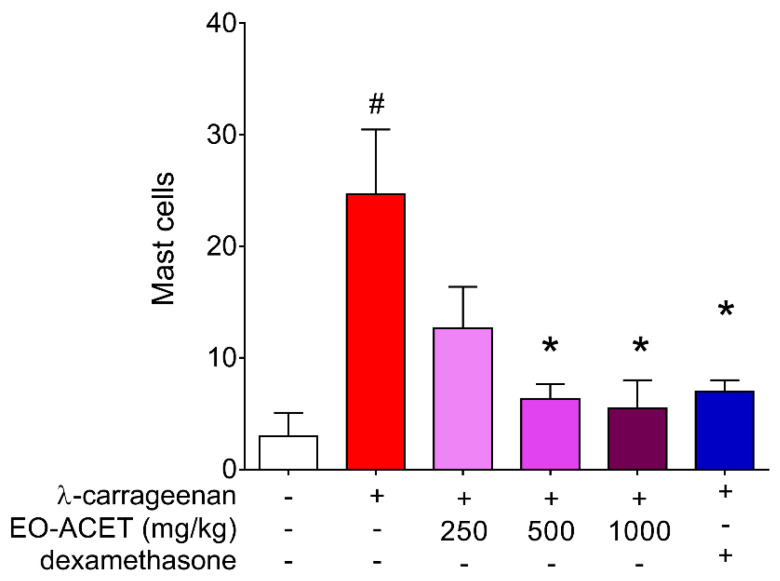
Mast cells counting in BALB/c mice footpad inoculated with of λ-carrageenan and treated with ethyl acetate fraction of *Euterpe oleraceae* (EO-ACET). Animals inoculated with 25 μL of λ-carrageenan 1% and treated with 100 μL EO-ACET (purple columns) by gavage or with dexamethasone (blue column) at 5 mg/kg via the intramuscular route. Data represent the mean ± standard deviation of the experiment performed at least in triplicate. # *p* < 0.001 compared with the group without stimulation or treatment (white column); * *p* < 0.05, when compared with the stimulated and untreated group (red column) after Kruskal–Wallis analysis followed by Dunn’s multiple comparisons tests.

**Table 1 foods-10-01014-t001:** Constituents found in *Euterpe oleraceae* seed extracts with their respective molecular pseudo-ions, relative percentage areas and their tentative identification.

Peak	RT ^1^	*m/z* (M+H^+^)	EO-HE ^2^	EO-CLO ^3^	EO-ACET ^4^	EO-AQ ^5^	Identification ^6^
1	1.1	203.0528	16.07	25.22	17.06	43.07	Not identified
2	1.3	231.0845	33.87	32.31	15.32	46.07	Not identified
3	1.6	231.0841	9.61	18.06	3.65	10.84	Not identified
4	2.8	867.2142	1.70	-	7.66	-	trimeric procyanidins
5	3.0	579.1488	10.23	-	19.62	-	pelargonidin-3-rutinoside
6	3.7	291.0860	25.24	21.40	30.91	-	Catechin
7	4.6	291.0855	3.25	2.98	5.75	-	Epicatechin

^1^ Retention time. ^2^
*E. oleracea* hydroalcoholic extract. ^3^
*E. oleracea* chloroformic fraction. ^4^
*E. oleracea* ethyl acetate fraction. ^5^
*E. oleracea* aqueous fraction. ^6^ Tentative identification.

**Table 2 foods-10-01014-t002:** Thickness in millimeters and percentage of λ-carrageenan 1% paw edema inhibition in BALB/c mice treated with ethyl acetate fraction of *Euterpe oleraceae*.

Treatment	λ-Carragenan	Dose (mg/kg)	Administration Time (% Edema Inhibition)			
			1 h	2 h	3 h	4 h
PBS	-	-	0.04 ± 0.055	0.02 ± 0.045	0.02 ± 0.045	– ^1^
	+	-	0.22 ± 0.084	0.32 ± 0.134	0.48 ± 0.084	0.56 ± 0.055
EO-ACET	+	250	0.20 ± 0.100	0.25 ± 0.129	0.36 ± 0.089 (25.00)	0.46 ± 0.089 (17.85)
	+	500	0.22 ± 0.096	0.25 ± 0.058	0.34 ± 0.114 (29.16)	0.36 ± 0.114 (35.71) *
	+	1000	0.20 ± 0.100	0.25 ± 0.058	0.32 ± 0.084 (33.33)	0.34 ± 0.055 (39.29) **
Dexamethanose	+	5	0.18 ± 0.084	0.20 ± 0.100	0.20 ± 0.071 (58.33) ****	0.20 ± 0.071 (64.28) ****

Data represent the mean ± standard deviation of the experiment performed in quintuplicate. ^1^ There was no edema thickness in this group at this evaluation time. * *p* < 0.05; ** *p* < 0.01; **** *p* < 0.0001, after analysis of variance (two-way ANOVA) followed by Bonferroni’s multiple comparisons test when compared to the PBS group. EO-ACET: *Euterpe oleracea* ethyl acetate fraction. +: λ-carrageenan inoculation; -: PBS administration.

## Data Availability

All datasets presented in this study are included in the article.

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
