# Peer review of "Inhibitory Effect of Catechin-Rich Açaí Seed Extract on LPS-Stimulated RAW 264.7 Cells and Carrageenan-Induced Paw Edema"

_foods, 2021, doi:10.3390/foods10051014_

Round 1
Reviewer 1 Report
- The manuscript entlited “Inhibitory effect of catechin-rich açaí seed extract on LPS-stimulated RAW 264.7 cells and carrageenan-induced paw edema”, authored by Gabriel Silva Xavier and colleagues, deals with the evaluation of the effect of catechin-rich açaí extract on LPS-stimulated RAW 264.7 cells and carrageenan-induced paw edema. This article is very interesting from a scientific point of view, however some small corrections should be made.
- Introduction should be improved with more botanical information about Euterpe Oleracea and with previous literature data regarding antioxidant properties of phenolic compounds. The methodology should be described if it was reported in the papaer or the authors should include related reference.
- Topic is original and it offers a first evaluation of in vitro and in vivo anti-inflammatory activity induced by EO-ACET treatment.
- The paper is well written and text is clear to read.
- The conclusions are consistent with the results, but the authors should complement the discussion with previous literature data regarding the anti-inflammatory markers evaluated.
- The authors should introduce a further section to conclusions only and they shouldn’t include conclusions with the discussion.
- The reported acronymus should be explain in extenso at their first appearance in the text: LPS, IL 1-β, IL-6, IL-12, NLRP3, MAPK, TLR4, TLR2/4 NF-kB.
- Line 138: it should be 3-(4,5-dimetyl-2-yl)-2,5-diphenyltetrazollium bromide (MTT)
- Section 2.5: The reference for determination Quantification of Endotoxins is missing.
Author Response
Dear reviewer,
We thank very much for your valuables comments, and for the time spent in our manuscript.
The manuscript has now been thoroughly revised in the light of very comments and suggestions and changes made are indicated in the revised manuscript highlighted by "Track Changes" function in Microsoft Word.
Kind regards,
Kátia da Silva Calabrese.
Reviewer 1
- The manuscript entlited “Inhibitory effect of catechin-rich açaí seed extract on LPS-stimulated RAW 264.7 cells and carrageenan-induced paw edema”, authored by Gabriel Silva Xavier and colleagues, deals with the evaluation of the effect of catechin-rich açaí extract on LPS-stimulated RAW 264.7 cells and carrageenan-induced paw edema. This article is very interesting from a scientific point of view, however some small corrections should be made.
Answer: We made all corrections requested by reviewer.
- Introduction should be improved with more botanical information about Euterpe Oleracea and with previous literature data regarding antioxidant properties of phenolic compounds. The methodology should be described if it was reported in the papaer or the authors should include related reference.
A: Botanical information (lines 71-81), including a figure of tree, fruits and seeds of E. oleracea (Figure 1), and antioxidant properties of phenolic compounds (lines 82-94) were added to Introduction section. Methodologies were revised and references were added to less detailed methods as requested.
- Topic is original and it offers a first evaluation of in vitro and in vivo anti-inflammatory activity induced by EO-ACET treatment.
A: We thank you for your evaluation
- The paper is well written and text is clear to read.
A: We thank you for your comment.
- The conclusions are consistent with the results, but the authors should complement the discussion with previous literature data regarding the anti-inflammatory markers evaluated.
A: We added more discussion with previous literature data as requested (Lines 391-398, 416-419)
- The authors should introduce a further section to conclusions only and they shouldn’t include conclusions with the discussion.
A: We added a Conclusions as requested.
- The reported acronymus should be explain in extenso at their first appearance in the text: LPS, IL 1-β, IL-6, IL-12, NLRP3, MAPK, TLR4, TLR2/4 NF-kB.
A: All acronyms were explained at first appearance.
- Line 138: it should be 3-(4,5-dimetyl-2-yl)-2,5-diphenyltetrazollium bromide (MTT)
A: Corrected in the manuscript.
- Section 2.5: The reference for determination Quantification of Endotoxins is missing.
A: Reference to endotoxin quantification was added to the manuscript (Line 176).
Reviewer 2 Report
The manuscript submitted by Siva Xavier et al., described how E. oleracea seeds extract exert antinflammatory effects both in vitro in RAW264.7 cells and in an in vivo model of acute inflammation.
The manuscript is well described and reults are consitent with data. Anyhow some changes are needed in order to improved the work.
1.- English need to be corrected.
2.- When do in vitro work, authors only used supernatant to check inflammation. what did they do with cells? Experiments in cell lisate are needed in order to adscribe a molecular pathway involved in inflammation.
3.- When describe Paw edema methodology: ´The edema thickness was expressed in millimeters and obtained from the difference between the inoculated footpad and the non-inoculated footpad´. It is more acurate to compare inoculated footpad with its own basal level. The data should be redescribed according to this.
4.- Images from in vivo model (paw edema) are important to understand data in table and figure. The same for mast cells number.
Author Response
Dear reviewer,
We thank very much for your valuables comments and for the time spent in our manuscript.
The manuscript has now been thoroughly revised in the light of very comments and suggestions and changes made are indicated in the revised manuscript highlighted by "Track Changes" function in Microsoft Word.
Kind regards,
Kátia da Silva Calabrese
Reviewer 2
The manuscript submitted by Siva Xavier et al., described how E. oleracea seeds extract exert antinflammatory effects both in vitro in RAW264.7 cells and in an in vivo model of acute inflammation.
The manuscript is well described and reults are consitent with data. Anyhow some changes are needed in order to improved the work.
1.- English need to be corrected.
Answer: The manuscript was checked by the professional English editing service of Prof. James Yong, an English native speaker, before submission. The revised version of the manuscript was also verified.
2.- When do in vitro work, authors only used supernatant to check inflammation. what did they do with cells? Experiments in cell lisate are needed in order to adscribe a molecular pathway involved in inflammation.
A: We agree with you. Initially, we kept the cells frozen at -80 ºC to perform molecular studies of gene expression and protein quantification and analyze inflammation pathways. However, we need financial support for this kind of study, which was reduced due to COVID-19 pandemic. Currently, we are looking for financial assistance to carry out these molecular experiments, as well as other in vivo inflammation models. As this is the first report of anti-inflammatory activity of açaí seed, and considering that the results obtained are consistent, we chose to finalize this paper with these results and proceed with the other evaluations in further studies as the funding is released. Anyway, we believe that manuscript presents relevant results and encourages us to carry out these studies in the very near future.
3.- When describe Paw edema methodology: ´The edema thickness was expressed in millimeters and obtained from the difference between the inoculated footpad and the non-inoculated footpad´. It is more acurate to compare inoculated footpad with its own basal level. The data should be redescribed according to this.
A: We revised the data and described paw edema methodology according reviewer request.
4.- Images from in vivo model (paw edema) are important to understand data in table and figure. The same for mast cells number.
A: Images were added to the manuscript as requested, see Figure 5 and Figure 6.
Reviewer 3 Report
The manuscript entlited “Inhibitory effect of catechin-rich açaí seed extract on LPS-stimulated RAW 264.7 cells and carrageenan-induced paw edema”, authored by Gabriel Silva Xavier and colleagues, deals with the evaluation of the effect of catechin-rich açaí extract on LPS-stimulated RAW 264.7 cells and carrageenan-induced paw edema. I think the article is of particular scientific interest, however some observations need to be made.
- In the introduction, some botanical information regarding the Euterpe oleracea should be added.
- In the introduction, some information regarding the anti-inflammatory activity of other common berries should be added (ex. https://doi.org/10.3390/antiox8080299; https://doi.org/10.3390/molecules22020312; https://doi.org/10.3390/molecules23071812).
- In the introduction section, the authors should underline that in the last two years an increase of consumer interest in foods with high nutraceutical, as well as nutritional value, has been documented. In particular, many consumers believe that the consumption of foods rich in particular phytochemicals can be useful and have beneficial effects on human health, both for prevention and for the treatment of specific pathologies. This idea followed in the functionalization of foods and drinks with fruit-based extracts that claim to produce several and different beneficial effects on human well-being. Moreover, this also strongly effected the food supplement market, which has documented an exponential growth in the production and sale of dietary supplements, especially those berry-based (https://doi.org/10.3390/nu12040992; https://doi.org/10.3390/nu13010054). Indeed, epidemiological studies have already shown that the intake of supplements based on berry fruits may have several positive and profound effects on human health (https://doi.org/10.3390/nu12040992; https://doi.org/10.3109/13880209.2012.674141). Please, report these references and if necessary add more.
- (optional) I understand that the authors' main target is the evaluation of the anti-inflammatory activity of the extracts, and not their chemical characterization. However, a quantification of the bioactive compounds could be interesting in the manuscript.
- The figures should be improved in quality. In particular, since does not apply additional costs for the publication of coloured images, I recommend to resubmit a coloured version of the figures.
- In table 2, all the values should report the same significative figures (0.04 ± 0.05). Moreover, 0.00±0.000 does have any sense. Please, remove it and change it with ‘-‘.
- Line 206: how can authors state that “a small amount of pelargonidin was identified” if quantitative analyses were not performed? The relative area percentage resulting from a MS/MS analysis cannot be used to make such considerations.
- The discussion section should be supplemented by comparisons with previous literature data, in which the same biological parameters have been evaluated.
- I recommend to add an additional section for the conclusions.
Author Response
Dear reviewer,
We thank very much for your valuables comments and for the time spent in our manuscript.
The manuscript has now been thoroughly revised in the light of very comments and suggestions and changes made are indicated in the revised manuscript highlighted by "Track Changes" function in Microsoft Word.
Kind regards,
Kátia da Silva Calabrese
Reviewer 3
The manuscript entlited “Inhibitory effect of catechin-rich açaí seed extract on LPS-stimulated RAW 264.7 cells and carrageenan-induced paw edema”, authored by Gabriel Silva Xavier and colleagues, deals with the evaluation of the effect of catechin-rich açaí extract on LPS-stimulated RAW 264.7 cells and carrageenan-induced paw edema. I think the article is of particular scientific interest, however some observations need to be made.
- In the introduction, some botanical information regarding the Euterpe oleracea should be added.
Answer: Botanical information was added to Introduction section (Line 71-87), including a figure of tree, fruits and seeds of E. oleracea (Figure 1).
- In the introduction, some information regarding the anti-inflammatory activity of other common berries should be added (ex. https://doi.org/10.3390/antiox8080299; https://doi.org/10.3390/molecules22020312; https://doi.org/10.3390/molecules23071812).
A: We added the references to the Introduction section (Lines 64-70).
- In the introduction section, the authors should underline that in the last two years an increase of consumer interest in foods with high nutraceutical, as well as nutritional value, has been documented. In particular, many consumers believe that the consumption of foods rich in particular phytochemicals can be useful and have beneficial effects on human health, both for prevention and for the treatment of specific pathologies. This idea followed in the functionalization of foods and drinks with fruit-based extracts that claim to produce several and different beneficial effects on human well-being. Moreover, this also strongly effected the food supplement market, which has documented an exponential growth in the production and sale of dietary supplements, especially those berry-based (https://doi.org/10.3390/nu12040992; https://doi.org/10.3390/nu13010054). Indeed, epidemiological studies have already shown that the intake of supplements based on berry fruits may have several positive and profound effects on human health (https://doi.org/10.3390/nu12040992; https://doi.org/10.3109/13880209.2012.674141). Please, report these references and if necessary add more.
A: We thank you and added your reference suggestions to the Introduction section (Lines 39-63).
- (optional) I understand that the authors' main target is the evaluation of the anti-inflammatory activity of the extracts, and not their chemical characterization. However, a quantification of the bioactive compounds could be interesting in the manuscript.
A: We agreed with you but unfortunately due to COVID-19 pandemic we are not able to perform the quantification of the bioactive compounds in time for this paper. We hope to be able to add this results in further studies, and we believe that the chemical characterization presented in the manuscript is a reliable evaluation that will direct our and other future studies.
- The figures should be improved in quality. In particular, since does not apply additional costs for the publication of coloured images, I recommend to resubmit a coloured version of the figures.
A: We appreciate your suggestion and add color to the figures. The figures in the manuscript are at least 300 dpi.
- In table 2, all the values should report the same significative figures (0.04 ± 0.05). Moreover, 0.00±0.000 does have any sense. Please, remove it and change it with ‘-‘.
A: We corrected and added footnote in Table 2.
- Line 206: how can authors state that “a small amount of pelargonidin was identified” if quantitative analyses were not performed? The relative area percentage resulting from a MS/MS analysis cannot be used to make such considerations.
A: Corrected in the manuscript (Line 340).
- The discussion section should be supplemented by comparisons with previous literature data, in which the same biological parameters have been evaluated.
A: Discussion section were supplemented with previous literature data (Lines 349-356, 391-398, 416-419)
- I recommend to add an additional section for the conclusions.
A: Conclusions section was added to the manuscript.
Round 2
Reviewer 2 Report
Authors have answered all my questions. I don´t understand the new Covid-19 paragraph in the introduction section page 2 lines 50-65. I think does not below to the described topic and it can be deleted.
Author Response
Dear Reviewer, we appreciate and agree with your comment regarding the paragraph on COVID. It was added at the request of another reviewer, who listed the references of the paragraph mentioned. The manuscript was submitted with the paragraph removed.